# Impact of COVID-19 on Adherence to Treatment in Patients with HIV

**DOI:** 10.3390/healthcare11091299

**Published:** 2023-05-02

**Authors:** Pablo Carbonero-Lechuga, Javier Castrodeza-Sanz, Iván Sanz-Muñoz, Pilar Marqués-Sánchez, Jose M. Eiros, Carlos Dueñas-Gutiérrez, Camino Prada-García

**Affiliations:** 1Department of Preventive Medicine and Public Health, University of Valladolid, 47005 Valladolid, Spain; pclcarbonero@gmail.com (P.C.-L.);; 2Preventive Medicine and Public Health Service, Hospital Clínico Universitario de Valladolid, 47003 Valladolid, Spain; 3National Influenza Centre, Edificio Rondilla, Hospital Clínico Universitario de Valladolid, 47009 Valladolid, Spain; 4SALBIS Research Group, Faculty of Health Sciences, Ponferrada Campus, Universidad de León, 24401 Ponferrada, Spain; 5Microbiology Service, Hospital Universitario Río Hortega, 47012 Valladolid, Spain; 6Infectious Diseases Unit, Hospital Clínico Universitario, 47003 Valladolid, Spain; 7Dermatology Service, Complejo Asistencial Universitario de León, 24008 León, Spain

**Keywords:** HIV, adherence, COVID-19, antiretroviral treatments, SMAQ

## Abstract

In patients with human immunodeficiency virus (HIV), adherence to treatment is affected by the adverse effects of treatment, the presence of additional comorbidities, the complexity of dosage, and family and community support. However, one recent circumstance that was likely to have influenced therapeutic adherence was the COVID-19 pandemic and the applied containment measures. An observational retrospective study of a sample of patients with HIV was conducted to establish the relationship between sociodemographic, clinical, and pharmacological variables and therapeutic adherence before and after the pandemic. Adherence was measured using the validated simplified medication adherence questionnaire (SMAQ) and medication possession rate. A statistical analysis was performed to determine the mean, standard deviation, and median of the quantitative variables and the frequencies of the qualitative variables, and the relationship between the dependent and independent variables was analysed using the chi-squared test and Student’s *t*-test. No statistically significant differences were found between treatment adherence measured before and 22 months after the start of the pandemic. Sex, occupation, treatment regimen, viral load levels, and COVID-19 disease status did not influence adherence during either period. However, the age of patients with HIV had an impact on adherence during both periods (*p* = 0.008 and *p* = 0.002, respectively), with the age group under 45 years being less adherent. In addition, experiencing adverse drug reactions (ADRs) was shown to have an impact on adherence before the pandemic (*p* = 0.006) but not afterwards. The COVID-19 pandemic was not shown to have an impact on the degree of adherence to antiretroviral treatment in patients with HIV. Instead, adherence was influenced by patient age and ADR occurrence; therefore, measures must be taken in this regard. The SMAQ demonstrated sensitivity in assessing adherence.

## 1. Introduction

Adherence to antiretroviral therapy has been extensively studied since the introduction of the first drugs in the fight against human immunodeficiency virus (HIV). The importance of good treatment adherence for the quality of life and life expectancy of HIV-positive patients is indisputable [1]. Therefore, the determinants that favour or prevent patients from taking their medication have been the subject of research to improve health interventions among the most vulnerable groups. It has been shown that the presence of other comorbidities, especially a psychiatric pathology [2], the adverse effects that pharmaceuticals can produce [3], and the complexity of dosage [4] have an impact on adherence. From a psychosocial point of view, the stigma surrounding HIV [5], the absence of family or community support [6], and substance abuse [7] also influence treatment adherence. Additionally, the psychological wellbeing of sexual and gender minorities with HIV/AIDS or cancer has been shown to be significantly impacted, highlighting the importance of addressing the mental health needs of these vulnerable populations [8]. Healthcare systems and professionals within these systems [9] are also considered relevant factors for compliance, although their influence has not been studied as extensively as the other determinants mentioned above [10,11]. Therefore, it is clear that adherence is impacted by all spheres of life for HIV-positive individuals. Moreover, the success of antiretroviral treatment depends on many disparate factors, making the management of these patients challenging. 

Under these circumstances, both temporary and permanent societal conditions also play significant roles. If any recent single social circumstance has had a crosscutting influence on all spheres of life for an individual with HIV, it is undoubtedly the pandemic caused by COVID-19. Confinement and subsequent measures to contain the virus have created a turning point at both the social and individual levels. Interpersonal relations have changed, and how we use healthcare systems has become radically different, requiring more reliance on telephone consultations and the postponement of addressing health problems until a later time in some cases. While the pandemic has affected all of us in some ways, its impact on users of healthcare systems has likely been among the most significant. It is believed that chronically ill patients, specifically patients with HIV, have suffered the most from the consequences of pandemic measures, as they have been unable to regularly use health services [12]. 

COVID-19 outcomes and risk factors among people living with HIV have been reported, with some studies suggesting that HIV-positive patients might be at a higher risk of complications, not only because of immunosuppression but also because of any interruption to their disease monitoring as well as social isolation [13]. Furthermore, the efficacy of COVID-19 vaccines in immunocompromised patients, including those with HIV, has been a topic of interest, with most studies suggesting that there are no significant differences in immune responses between HIV-positive and HIV-negative patients [13,14,15]. In summary, HIV-positive patients have experienced unique vulnerabilities during the pandemic. Heterologous prime-boost strategies for COVID-19 vaccines have also been explored in the context of HIV and other immunocompromised populations, offering the potential for more flexible and robust vaccination schedules [16]. Additionally, research on the safety, immunogenicity, and effectiveness of COVID-19 vaccines in special populations, such as pregnant and breastfeeding women and their infants, has also provided valuable information to guide clinical practice and public health policy [17].

In summary, HIV-positive patients have experienced unique vulnerabilities during the pandemic, and the impact of COVID-19 on adherence to antiretroviral therapy is a critical area of research. Taking all of these into account, we can infer that adherence to treatment in patients with HIV might have been affected by the surrounding pandemic-related changes. Therefore, the primary objective of this study was to analyse the impact that the pandemic caused by SARS-CoV-2 had on therapeutic adherence to antiretroviral drugs in a selected group of HIV-positive patients at the Hospital Clínico Universitario de Valladolid (HCUV). We also sought to clarify which variables had the greatest influence on treatment adherence in our patients to mitigate the effects that any similar circumstances may produce in the future. 

However, assessing adherence to treatment can be challenging because of the nature of its various determinants, and this was a crucial aspect during the conduct of this study. Following the recommendations of the Grupo de Estudio de (GeSIDA, an AIDS Study Group) and using a combination of indirect methods for measuring adherence, we were able to differentiate between adherent and non-adherent patients while simultaneously assessing our measurement methods ourselves, observing any discrepancies in the results regarding each of the specific methods.

This study aimed not only to contribute to the growing body of literature on the impact of the COVID-19 pandemic on HIV-positive patients but also to inform clinical practice and public health policy in addressing the unique challenges faced by this population during such an unprecedented global health crisis. By examining the factors influencing adherence to antiretroviral therapy and incorporating the findings from recent studies on COVID-19 outcomes, risk factors, and vaccine efficacy in HIV-infected patients [13,14,15], we aimed to provide a comprehensive understanding of the interplay between the COVID-19 pandemic and HIV treatment adherence. Furthermore, the exploration of heterologous prime-boost strategies [16] and the safety and effectiveness of COVID-19 vaccines in special populations [17] added valuable insights into potential strategies for optimising vaccination efforts among HIV-positive patients and other vulnerable groups.

In summary, this study sought to elucidate the impact of the COVID-19 pandemic on adherence to antiretroviral therapy among HIV-positive patients, identify the variables with the greatest influence on treatment adherence, and inform future interventions to mitigate the effects of similar circumstances on this vulnerable population. By incorporating findings from recent studies on COVID-19 outcomes, risk factors, and vaccine efficacy in HIV-infected patients and other special populations, we aimed to provide a comprehensive understanding of the complex relationship between the COVID-19 pandemic and HIV treatment adherence, ultimately contributing to the development of more effective strategies for managing and supporting HIV-positive patients during challenging times.

The manuscript is organised as follows: Section 2 provides a comprehensive overview of the research design, including subsections on the study design and scope (Section 2.1), inclusion and exclusion criteria (Section 2.2), variables (Section 2.3), data collection (Section 2.4), ethical aspects (Section 2.5), and statistical analysis (Section 2.6). Section 3 presents the results, with a description of the sample (Section 3.1) and an analysis of adherence (Section 3.2). Section 4 covers the discussion, where the findings are analysed and contextualised. Finally, Section 5 offers the conclusions, summarising the key takeaways and their implications for the field.

## 2. Material and Methods

### 2.1. Study Design and Scope

This study was based on a longitudinal and retrospective observational study of patients hospitalised in the HCUV for complications arising from HIV infection between 1 January 2010 and 31 December 2019. After applying the inclusion and exclusion criteria described below, 43 patients were included in the study. The data were collected between December 2021 and February 2022.

### 2.2. Inclusion and Exclusion Criteria

The inclusion criteria used in this study were as follows:Living patients who were over the age of majority at the time of the interview;Patients diagnosed with HIV before February 2019;Patients who had attended at least one telephone or face-to-face consultation between May 2020 and November 2021;Patients with at least two viral load determinations: one before February 2020 and one after June 2021.

The exclusion criteria were as follows:Patients who did not meet the inclusion criteria;Patients who had missing data for any of the variables analysed in the study;Patients who had less than two viral load determinations during the study period.

### 2.3. Variables

The variables measured included age; sex; occupation; home-based support; the initial viral load determination before February 2020; the last viral load determination since June 2021; the number of antiretroviral treatment (ART) tablets per day; daily medication intake; the presence or absence of adverse drug reactions to ART; the presence of at least one positive test result (polymerase chain reaction or antigen test) for SARS-CoV-2; adherence to ART before February 2020 using the validated simplified medication adherence questionnaire (SMAQ); medication possession rate from March 2018 to February 2022; adherence to ART in November 2021 using the validated SMAQ; and medication possession rate between March 2020 and February 2022. The SMAQ, which has been validated in the Spanish population [18], aims to assess adherence and consists of six dichotomous response questions, in which at least one ‘non-adherent’ response is considered as indicating that the patient is non-adherent. The rate of medication possession, which is ultimately the rate of attendance at the hospital pharmacy service, was calculated as the quotient between the number of total galenic units dispensed and the number expected, considering a patient to be non-adherent if the possession rate was ≤80% per the protocols used by the HCUV Pharmacy Service. Both the SMAQ and the medication possession rate are considered indirect methods of measuring adherence according to the guidelines established by the GeSIDA group, which considers a combination of both as providing the minimum acceptable information for assessing adherence to ART in people with HIV [19].

### 2.4. Data Collection

Age, sex, viral load at two time points, and SARS-CoV-2 diagnostic test results were extracted from medical histories. Information on home-based support, occupation, ART tablets per day, daily medication intake, presence of adverse reactions to ART, and degree of adherence at two points in time using the SMAQ was collected through a standardised personal interview via telephone. After the objectives and general aspects of the study were provided, informed consent and explicit acceptance by the patients to participate in the study were obtained. Using the pharmacological dispensing register offered by the Pharmacology Service of the HCUV through the Farmatools^®^ program, the rate of medication possession during the two periods was calculated.

All the data described above were collected using an Excel table. The viral load was measured in copies/mL. However, in this study, we only had access to the data on whether or not the viral load was detectable, not the exact value of the load. To preserve the anonymity of the patients, each patient was assigned a random number, and the link to the patient’s medical record number was kept on one of the hospital computers under password protection and subsequently deleted.

### 2.5. Ethical Aspects

Data collection for this study began after the approval of the study by the Clinical Research Ethics Committee of the HCUV, which approved the study on 9 December 2021 based on the agreements and regulations of the Spanish legislation regarding protected personal and bioethical data. All patients who agreed to participate voluntarily in the study were informed of the objectives of the study, the anonymous nature of their personal data and responses, and the revocability of their consent at any time during and after the interview.

### 2.6. Statistical Analysis

Statistical analysis was performed using SPSS^®^ software version 27.0.1.0 (IBM Corp., Armonk, NY, USA). The descriptive part of this analysis was conducted by determining the mean, standard deviation, and median of the quantitative variables and the frequencies of the qualitative variables. The analysis was conducted by considering adherence before the start of the pandemic and adherence at 22 months after the start of the pandemic as the independent variables. The relationship between the dependent and independent variables was analysed using Pearson’s chi-squared test (χ^2^) for qualitative variables and the Student’s *t*-test for quantitative variables. The significance level was set at *p* < 0.05.

## 3. Results

### 3.1. Sample Description

The study sample consisted of 25 individuals with an average age of 51.7 years (SD = 10.7) and a median age of 52 years. The age range was divided into 3 categories: 6 (24%) participants were under 45 years old, 13 (52%) were between 45 and 60 years old, and 6 (24%) were over 60 years old. The majority of participants were male (76%), and their professional status was diverse, with 10 (40%) active workers, 4 (16%) unemployed individuals, and 11 (44%) retired individuals. Most of the participants lived with someone else (68%), while eight (32%) lived alone. Before COVID-19, 9 (36%) participants had detectable viral loads, while 16 (64%) had non-detectable viral loads. After COVID-19, 6 (24%) participants had detectable viral loads, while 19 (76%) had non-detectable viral loads. Twenty (80%) participants had reached the AIDS stage, while five (20%) did not. The average number of tablets per day was 1.5 (SD = 0.7). Further socio-demographic data and data on the medical study variables are shown in Table 1.

Adherence to antiretroviral therapy was analysed using the SMAQ and the HCUV drug-dispensing register, both validated for measuring adherence in patients with HIV [19], at two points in time: before February 2020 and in November 2021. Large discrepancies between the results obtained using both methods were observed after cross-checking them (Table 2 and Table 3).

### 3.2. Adherence Analysis

The study variables were cross-referenced with the adherence data obtained from the SMAQ both before and after the pandemic using Pearson’s chi-squared test (χ2) for the qualitative variables and the Student’s t-distribution for the quantitative variables (Table 4 and Table 5). Age was shown to be a determinant of the degree of adherence in the participating individuals, both before (*p* = 0.008) and after the pandemic (*p* = 0.002). The presence of adverse drug reactions (ADR) had a statistically significant influence on adherence before the pandemic (*p* = 0.043) but not after the pandemic (*p* = 0.244). The rest of the variables in this study did not show a statistically significant impact, either before or after the pandemic.

The difference between adherence before and after the pandemic was not observed to be significant, with only one patient going from being non-adherent to adherent.

## 4. Discussion

This study showed that the pandemic had no impact on patients’ adherence to antiretroviral therapy for HIV. Furthermore, neither the dosage of antiretroviral treatment nor experiencing COVID-19 illness had any effect on adherence. Age < 45 years did influence adherence both before and after the pandemic. Adverse drug reactions (ADR) to antiretroviral treatment impacted adherence after the pandemic but not beforehand.

Measuring adherence remains a challenge in clinical practice as existing methods are relatively reliable but tend to overestimate or underestimate actual adherence values. In our study, we calculated the medication possession rate using a drug-dispensing register. We found that 87.5% of patients who were previously non-adherent according to the SMAQ became adherent before the pandemic. Furthermore, all non-adherent patients were considered adherent after the pandemic. Although clinical practice tends to overestimate adherence [20], and the two methods used in this study have this issue of overestimation as a disadvantage in particular [21], it is remarkable that the data from the drug-dispensing registry differ significantly from the adherence calculated from the questionnaire. Such a marked overestimation of adherence by the drug-dispensing register may lead to errors in assessing patients as adherent, and the implementation of different interventions to improve adherence may be compromised. Therefore, during this study, the drug-dispensing register was considered to be a more overestimated measure of adherence than the validated SMAQ, and adherence was, therefore, analysed based on the results obtained by the SMAQ.

Age, above all the other variables studied, was shown to have had a sustained impact throughout the pandemic on adherence to treatment in patients with HIV. Younger people had lower adherence rates than older people did, which might be explained by the greater amount of experience that older patients have with the disease, having adapted the treatment to fit into their daily lives. Previous work in this area has concluded that age is a key determinant of adherence [22,23,24] and that young people are precisely the group that should be targeted by health interventions to improve adherence rates. Generally, older adults tend to have a better adherence to ART compared to younger individuals. Some factors contributing to this difference include life experience, maturity, and better coping skills among older adults. Therefore, it is necessary to redirect these interventions, which, once the group of patients most vulnerable to adherence has been identified, must be adapted as necessary. It was observed that adverse drug reactions (ADRs) were a significant factor contributing to non-adherence to HIV treatments before the COVID-19 pandemic. However, their significance diminished after the pandemic [25]. This change could be attributed to the heightened awareness and fear surrounding COVID-19, leading individuals to tolerate ADRs to maintain their overall health and reduce their vulnerability to the virus. People living with HIV might have adapted to the challenges posed by the pandemic to maintain their health, as suggested in [26], but further research is needed to understand the specific reasons behind this shift in treatment adherence and the role of ADRs in this context. Patients with poorer treatment experiences are expected to have lower adherence rates due to drug reluctance or distrust. Despite the low rates of adverse effects of current ART, patients should continue to be educated about the possibility of adverse effects, which are often presented as normal and temporary effects of the drugs or which can be easily resolved by changing medication. In contrast, the relationship between the gender of HIV-positive individuals and their treatment adherence is not well defined [27], as observed in this study. This may indicate that men and women are adherent and non-adherent in similar proportions, although the ways in which they are adherent may differ. Similarly, the employability status of patients and its impact on adherence remain undefined, although previous studies determining the influence of this factor on individual adherence [28] found that adherence rates were higher in employed and working individuals. For HIV, as for many other diseases [29,30,31], social and family support are important determinants of adherence. Although we found no significant differences between those living alone and those living with someone else, it is not surprising that other studies point to a plausible influence on adherence, likely because living alone or with other people is not synonymous with support. Our study may differ from other studies, potentially due to the limited number of patients included in our sample. Although viral load is not a good indicator of adherence and does not provide information on adherence patterns among patients [32], it may be considered useful as an indirect method to make an initial assessment. Thus, although our study did not show a significant relationship between adherence and viral load, the latter did provide some remarkable information, such as the fact that 12.5% of patients who were non-adherent at the beginning of the pandemic became adherent after the pandemic. The complexity of antiretroviral treatment (mainly dependent on the number of tablets, doses, and dietary conditions) has a significant impact on adherence to such therapy, which has been repeatedly demonstrated [4,33]. However, we were unable to draw the same conclusions in our study, likely because of the current simplicity in the dosages prescribed to patients with HIV. 

Few studies have determined the impact of the SARS-CoV-2 pandemic on adherence to antiretroviral therapy [34,35]. This study found that the pandemic did not influence adherence among HIV-positive patients at the HCUV. However, more information on adherence is expected to be available in the future.

## 5. Conclusions

In conclusion, despite the impact of the pandemic on the mental health of patients with HIV at the HCUV and their perception of the healthcare system, our study demonstrated that it did not affect their adherence to antiretroviral treatment, regardless of gender or occupation. Neither antiretroviral treatment dosage nor COVID-19 disease status had an impact on adherence. In contrast, age, particularly < 45 years, influenced adherence both before and after the pandemic. Similarly, adverse drug reactions (ADRs) to antiretroviral treatment significantly impacted adherence before the pandemic but not afterwards. It is worth noting that this study found the SMAQ to be a highly valuable tool for assessing patients’ adherence to treatment. However, adherent patients persisted during the worst months of the pandemic. Implementing various interventions targeting the most vulnerable groups of HIV-positive patients should be a priority, not only during circumstances such as the COVID-19 pandemic but also on a day-to-day basis [36,37,38].

Our study has several limitations, including the limited sample size, the potentially insufficient follow-up period, the potential overestimation or underestimation of adherence measurements, and the possibility of unobserved variables influencing adherence. Future research should focus on addressing these limitations by conducting longitudinal studies to evaluate how adherence to antiretroviral treatment and associated factors evolve over time in patients with HIV, particularly in the context of the COVID-19 pandemic. Additionally, the effectiveness of specific interventions designed to improve adherence in vulnerable patient groups, such as younger individuals or those experiencing ADRs, should be investigated. Comparing and validating different adherence measurement methods and exploring the impact of the pandemic on other aspects of HIV care are also important avenues for future research. Identifying and analysing additional protective and risk factors that may influence antiretroviral treatment adherence during and after the pandemic will further contribute to our understanding and inform targeted interventions.

## Figures and Tables

**Table 1 healthcare-11-01299-t001:** Sociodemographic and medical variables of the study sample.

	N (%)
**Average age (SD)** **Median**	51.7 (10.7) 52
**Age range** <45 45–60 >60	6 (24.0) 13 (52.0) 6 (24.0)
**Sex** Male Female	19 (76.0) 6 (24.0)
**Professional status** Active worker Unemployed Retired	10 (40.0) 4 (16.0) 11 (44.0)
**Family status** Live alone Live with someone else	8 (32.0) 17 (68.0)
**Pre-COVID-19 viral load** Detectable Not detectable	9 (36.0) 16 (64.0)
**Post-COVID-19 viral load** Detectable Not detectable	6 (24.0) 19 (76.0)
**AIDS stage** Yes No	20 (80.0) 5 (20.0)
**Adverse reactions to treatment** Yes No Don’t know	6 (24.0) 17 (68.0) 2 (8.0)
**SARS-CoV-2 infection** Yes No	3 (12.0) 22 (88.0)
	**N (SD)**
**Average number of tablets per day**	1.5 (0.7)
**Average number of intakes per day**	1.0 (0.2)

**Table 2 healthcare-11-01299-t002:** Adherence to antiretroviral treatment before February 2020.

		SMAQ (Pre)	
		Adherent N (%)	Non-Adherent N (%)
Rate of Medical Possession (Pre)	Adherent	17 (100%)	7 (87.5%)
	Non-adherent	-	1 (12.5%)

**Table 3 healthcare-11-01299-t003:** Adherence to antiretroviral treatment in November 2021.

		SMAQ (Post)	
		Adherent N (%)	Non-Adherent N (%)
Rate of Medical Possession (Post)	Adherent	18 (100.0%)	7 (100.0%)
	Non-adherent	-	-

**Table 4 healthcare-11-01299-t004:** Analysis of adherence to antiretroviral therapy pre-COVID-19 pandemic.

	Pre-COVID-19 Adherence		
	Yes N (%)	No N (%)	*p*-Value
**Age range** <45 45–60 >60	1 (16.7) 11 (84.6) 5 (83.3)	5 (83.3) 2 (15.4) 1 (16.7)	**0.008**
**Sex** Male Female	13 (68.4) 4 (66.7)	6 (31.6) 2 (33.3)	0.936
**Professional status** Active worker Unemployed Retired	7 (70.0) 2 (50.0) 8 (72.7)	3 (30.0) 2 (50.0) 3 (27.3)	0.695
**Family status** Live alone Live with someone else	6 (75.0) 11 (64.7)	2 (25.0) 6 (35.3)	0.607
**AIDS stage** Yes No	14 (70.0) 3 (60.0)	6 (30.0) 2 (40.0)	0.669
**Adverse reactions to treatment** Yes No Don’t know	2 (33.3) 14 (82.4) 1 (50.0)	4 (66.7) 3 (17.6) 1 (50.0)	**0.043**
	**N (SD)**	**N (SD)**	** *p* ** **-Value**
**Average number of tablets per day**	1.53 (0.80)	1.50 (0.75)	0.931
**Average number of intakes per day**	1.06 (0.24)	1.00 (0.00)	0.504

Statistically significant results (*p* < 0.05) are highlighted in bold.

**Table 5 healthcare-11-01299-t005:** Analysis of adherence to antiretroviral therapy post-COVID-19 pandemic.

	Post-COVID-19 Adherence		
	Yes N (%)	No N (%)	*p*-Value
**Age range** <45 45–60 >60	1 (16.7) 12 (92.3) 5 (83.3)	5 (83.3) 1 (7.7) 1 (16.7)	**0.002**
**Sex** Male Female	14 (73.7) 4 (66.7)	5 (26.3) 2 (33.3)	0.739
**Professional status** Active worker Unemployed Retired	8 (80.0) 2 (50.0) 8 (72.7)	2 (20.0) 2 (50.0) 3 (27.3)	0.527
**Family status** Live alone Live with someone else	6 (75.0) 12 (70.6)	2 (25.0) 5 (29.4)	0.819
**AIDS stage** Yes No	15 (75.0) 3 (60.0)	5 (25.0) 2 (40.0)	0.504
**Adverse reactions to treatment** Yes No Don’t know	3 (50.0) 14 (82.4) 1 (50.0)	3 (50.0) 3 (17.6) 1 (50.0)	0.244
	**N (SD)**	**N (SD)**	** *p* ** **-value**
**Average number of tablets per day**	1.50 (0.78)	1.57 (0.78)	0.840
**Average number of intakes per day**	1.06 (0.24)	1.00 (0.00)	0.544

Statistically significant results (*p* < 0.05) are highlighted in bold.

## Data Availability

Data are available from the corresponding author upon reasonable request.

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
