# Peer review of "Impact of COVID-19 on Adherence to Treatment in Patients with HIV"

_healthcare, 2023, doi:10.3390/healthcare11091299_

Round 1

Reviewer 2 Report

Thank you for submitting this interesting review of adherence changes to ART in PLWH during COVID-19.

Strengths

- Explanation of methods, definition of adherence possession rate and SMAQ questionnaire

- Inclusion and exclusion criteria appropriate and well thought out

Considerations for Adjustment / Change / Evaluation

- Recommend reviewing the whole paper for sentence structure and punctuation changes, ensuring all sentences are "full" sentences and tense is appropriate

- Table 1, Sex: recommend changing from "man, woman" to "male, female"

- Table 1, Viral Load: I appreciate listing if the viral load was detectable / undetectable, but more information could be appreciated (ex: definitions of detectable versus undetectable if using that nomenclature versus listing the mean or median viral load)

- Paragraph Following Table 1 (Lines 168-175): I find that this paragraph repeats some of the statistical information that was previously discussed in the statistical analysis section. I recommend sticking to just results if possible 

- Paragraph Following Table 3, Adherence Analysis Section: Similarly to the comment above, I find that this section repeats the information in the statistical analysis section as it relates to the listing of the statistical tests used.

- Paragraph Following Table 3, Adherence Analysis Section: please ensure that AMR has been appropriately defined prior to listing the abbreviation here

- Table 4 and 5, P Value Column: remove the header that you are listing N (%) as you are not doing that for P values

- Table 4 and 5, Yes/No Columns: I would move "N (%)" to the individual characteristics that are reported this way, since your tablets and intakes are not reported in this way (they are reported as "N, SD" according to what you have listed.

Lines 204 - 208: I understand what this sentence is saying, but it is worded in a confusing manner - I recommend reviewing and re-evaluating the sentence structure

Lines 260-261: not a complete sentence 

Reviewer 3 Report

It is a very interesting and promising paper regarding the possible actions that should be taken in order to increase the adherence of these patients. 

However, some points must be addressed in the paper:

1) First of all, an explanation regarding the inclusion criteria must be added in order to clarify better which patients were targeted. In addition, no exclusion criteria are mentioned in the section " Inclusion and exclusion criteria" so either a statement like "There were no exclusion criteria" or "Only the patients with inclusion criteria were selected" should be added or even a change of the title should be made.

2) In line 199 is the the word "Nor" indicating a new sentence? Otherwise, correct it as "nor".

3) In lines 239-241, authors claim that "Although we found no significant differences between those living alone and those living with others, it is not surprising that other studies point to a plausible influence on adherence, likely because living alone or with others is not synonymous with support.". How can authors explain the absence of this difference in their study? Is the low number of participants or the inclusion criteria the possible cause?

4) Authors, in lines 259-260, also claim that "we showed that the pandemic had no impact  on their adherence to antiretroviral treatment regardless of gender or occupation. Nor did, antiretroviral treatment dosage or COVID-19 disease status had an impact on adherence" . Authors should clarify the importance of their study regarding the COVID-19 pandemy. If the pandemy did not affect the adherence at all, why is the paper focused mostly on the impact of the COVID-19 and not being presented more generally?

Reviewer 4 Report

The objective of the study was to analyze the impact of the COVID-19 pandemic on therapeutic adherence to antiretroviral drugs in HIV-positive patients and to identify variables that influenced adherence. The study found that the pandemic did not significantly affect adherence, but age and adverse drug reactions had an impact on adherence before the pandemic. The study used the Simplified Medication Adherence Questionnaire (SMAQ) to assess adherence.

Q1 In the abstract section,  it does not provide specific details on the methods used or the results obtained, which would be useful for readers to understand the scope of the study. Overall, the abstract could benefit from more specific information on the study design and results to better convey the significance of the study. Suggest the authors to revise. 

Q2. Sugget to enrich the literature section 

-For the citation that develops more about older men and healthy aging and the HIV situation in the literature, I suggest adding the reference paper.

doi: 10.3389/fpubh.2022.912980

The articles listed are relevant to building a literature review on COVID-19 outcomes and risk factors, clinical course and outcomes, efficacy of COVID-19 vaccines, and COVID-19 vaccine use in immunocompromised patients. They provide insights and findings on the topic, which can be used to support the review and provide a comprehensive understanding of the subject matter. The articles can also serve as a reference for further research on the topic.

Spinelli MA, Jones BLH, Gandhi M. COVID-19 Outcomes and Risk Factors Among People Living with HIV. Curr HIV/AIDS Rep. 2022 Oct;19(5):425-432. doi: 10.1007/s11904-022-00618-w. Epub 2022 Aug 5. PMID: 35930187; PMCID: PMC9362624.

Barbera LK, Kamis KF, Rowan SE, Davis AJ, Shehata S, Carlson JJ, Johnson SC, Erlandson KM. HIV and COVID-19: review of clinical course and outcomes. HIV Res Clin Pract. 2021 Aug;22(4):102-118. doi: 10.1080/25787489.2021.1975608. Epub 2021 Sep 12. PMID: 34514963; PMCID: PMC8442751.

Lee ARYB, Wong SY, Chai LYA, Lee SC, Lee MX, Muthiah MD, Tay SH, Teo CB, Tan BKJ, Chan YH, Sundar R, Soon YY. Efficacy of covid-19 vaccines in immunocompromised patients: systematic review and meta-analysis. BMJ. 2022 Mar 2;376:e068632. doi: 10.1136/bmj-2021-068632. PMID: 35236664; PMCID: PMC8889026.

Hippisley-Cox J, Coupland CA, Mehta N, Keogh RH, Diaz-Ordaz K, Khunti K, Lyons RA, Kee F, Sheikh A, Rahman S, Valabhji J, Harrison EM, Sellen P, Haq N, Semple MG, Johnson PWM, Hayward A, Nguyen-Van-Tam JS. Risk prediction of covid-19 related death and hospital admission in adults after covid-19 vaccination: national prospective cohort study. BMJ. 2021 Sep 17;374:n2244. doi: 10.1136/bmj.n2244. Erratum in: BMJ. 2021 Sep 20;374:n2300. PMID: 34535466; PMCID: PMC8446717.

Fu W, Sivajohan B, McClymont E, Albert A, Elwood C, Ogilvie G, Money D. Systematic review of the safety, immunogenicity, and effectiveness of COVID-19 vaccines in pregnant and lactating individuals and their infants. Int J Gynaecol Obstet. 2022 Mar;156(3):406-417. doi: 10.1002/ijgo.14008. Epub 2021 Nov 13. PMID: 34735722; PMCID: PMC9087489.

Duly K, Farraye FA, Bhat S. COVID-19 vaccine use in immunocompromised patients: A commentary on evidence and recommendations. Am J Health Syst Pharm. 2022 Jan 5;79(2):63-71. doi: 10.1093/ajhp/zxab344. PMID: 34455440; PMCID: PMC8499782.

Sapkota B, Saud B, Shrestha R, Al-Fahad D, Sah R, Shrestha S, Rodriguez-Morales AJ. Heterologous prime-boost strategies for COVID-19 vaccines. J Travel Med. 2022 May 31;29(3):taab191. doi: 10.1093/jtm/taab191. PMID: 34918097; PMCID: PMC8754745.

Plummer MM, Pavia CS. COVID-19 Vaccines for HIV-Infected Patients. Viruses. 2021 Sep 22;13(10):1890. doi: 10.3390/v13101890. PMID: 34696319; PMCID: PMC8540182.

McLean G, Kamil J, Lee B, Moore P, Schulz TF, Muik A, Sahin U, Türeci Ö, Pather S. The Impact of Evolving SARS-CoV-2 Mutations and Variants on COVID-19 Vaccines. mBio. 2022 Apr 26;13(2):e0297921. doi: 10.1128/mbio.02979-21. Epub 2022 Mar 30. PMID: 35352979; PMCID: PMC9040821.

Udoakang AJ, Djomkam Zune AL, Tapela K, Nganyewo NN, Olisaka FN, Anyigba CA, Tawiah-Eshun S, Owusu IA, Paemka L, Awandare GA, Quashie PK. The COVID-19, tuberculosis and HIV/AIDS: Ménage à Trois. Front Immunol. 2023 Jan 27;14:1104828. doi: 10.3389/fimmu.2023.1104828. PMID: 36776887; PMCID: PMC9911459.

Q3. Considering the correlation between COVID-19 and HIV patients is complex and multifaceted. The COVID-19 pandemic has had a significant impact on the healthcare systems and social interactions of individuals with HIV, leading to disruptions in access to care and potential exacerbation of comorbidities. Studies have shown that patients with HIV and COVID-19 may be at a higher risk of complications due to immunosuppression and interruption of disease monitoring, as well as social isolation. However, most studies have suggested that there are no differences in immune responses between HIV-positive and negative patients regarding COVID-19 vaccines. Adherence to antiretroviral treatment in patients with HIV may also have been affected by the pandemic-related changes, although a recent study found no statistically significant differences in treatment adherence before and after the pandemic. suggest the authors add the further research is needed to fully understand the impact of COVID-19 on patients with HIV and to develop effective strategies to mitigate any negative effects.

Q3. Appreciated. The data variable section of the study provides a comprehensive list of variables that were measured in the study. These variables cover various aspects of HIV treatment and adherence, including patient demographics (age, sex, occupation, household accompaniment), treatment regimen (number of ART tablets per day, daily medication intake, presence or absence of adverse drug reactions to ART), COVID-19 status (presence of at least one positive test result for SARS-CoV-2), and measures of adherence (adherence to ART before and after the pandemic using the SMAQ questionnaire, medication possession rate from March 2018 to February 2022, and medication possession rate between March 2020 and February 2022).

The use of the SMAQ questionnaire and medication possession rate as measures of adherence is consistent with the guidelines established by the GeSIDA group. The inclusion of both measures is also commendable as it provides a more comprehensive picture of adherence in patients with HIV.Overall, the data variable section is well-written and provides a clear description of the variables that were measured in the study.

Q4. Based on the information provided, the data collection methods used in this study appear to be appropriate and systematic. The data collected include both objective and subjective measures, such as medical records, interviews, and pharmacy records, which can provide a comprehensive view of medication adherence in HIV patients during the COVID-19 pandemic. Informed consent was also obtained from participants, and measures were taken to protect their privacy and anonymity. Therefore, the data collection process appears to be good. Suggest the authors provide the peiod of time for the data collection. 

Q5. For table 1 specifically highlighted, suggest the description below:

The study sample consisted of 25 individuals with an average age of 51.7 (SD=10.7) and a median age of 52. The age range was divided into three categories: 6 (24%) participants were under 45 years old, 13 (52%) were between 45 and 60 years old, and 6 (24%) were over 60 years old. The majority of participants were male (76%), and the professional status was diverse, with 10 (40%) active workers, 4 (16%) unemployed individuals, and 11 (44%) retired individuals. Most of the participants lived together with someone (68%), while 8 (32%) lived alone. Before COVID-19, 9 (36%) participants had detectable viral loads, while 16 (64%) had non-detectable viral loads. After COVID-19, 6 (24%) participants had detectable viral loads, while 19 (76%) had non-detectable viral loads. 20 (80%) participants had AIDS stage, while 5 (20%) did not. The average number of tablets per day was 1.5 (SD=0.7).

Q6. Suggest explaining the result in the results and the data analysis section. 

Q7. Suggest adding the limitation and the future implications section. 

Q8. Suggest submitting the editing service and checking the language quality; some wording in the manuscript is quite hard to read and understand. 

Round 2
